# Different but Synergistic Effects of Union and Manager Leadership on Member Job Satisfaction

**DOI:** 10.3390/bs14040287

**Published:** 2024-03-31

**Authors:** Heungjun Jung, Ki-Jung Kim, Inyong Shin

**Affiliations:** 1Department of Business Administration, Seoul National University of Science and Technology, Seoul 01811, Republic of Korea; hjunjung@seoultech.ac.kr; 2Department of Management, College of Business, Eastern Kentucky University, Richmond, KY 40475, USA; ki-jung.kim@eku.edu; 3Division of Business Administration, Pukyong National University, Busan 48513, Republic of Korea

**Keywords:** service-oriented union leadership, employee job satisfaction, union instrumentality, managerial ethical leadership

## Abstract

Existing research has tended to overlook the diverse roles of union leadership in contributing to member attitudes. Drawing on the social information processing theory, this study examines how union leaders’ (shop stewards) service-oriented leadership relates to member job satisfaction. To clarify the mechanism underlying this relationship, this study focuses on union instrumentality as a mediator. The research also examines managers’ ethical leadership as a conditional factor in the relationship between union leaders’ service-oriented leadership and member job satisfaction through union instrumentality. To test our hypothesis, this study analyzed the results of a survey of 603 respondents from two branches of the Korean Metal Workers’ Union. The findings of this study indicate that union instrumentality is the link between service-oriented union leadership and member job satisfaction. Additionally, the strength of the mediated relationship between the aforementioned factors through union instrumentality is contingent on managerial ethical leadership. This study contributes to an integrated understanding of the way in which service-oriented union stewards and ethical managers influence member job satisfaction through their leadership.

## 1. Introduction

Employee job satisfaction is a key determinant of a company’s competitive advantage in unionized workplaces, given that satisfied employees tend to have higher levels of engagement in the workplace and are more committed to their work, leading to better organizational performance. In unionized workplaces, unions and management constantly compete to win employees’ hearts and minds. Employee satisfaction even in the presence of unions positively impacts not only labor relations but also employees’ work attitudes, leading to organizational outcomes such as productivity and cooperation.

Because previous research has focused only on union leaders’ traditional roles and members’ attitudes towards labor unions, there is a lack of understanding of how union leadership influences member job attitudes and the mechanisms behind it. Research on the effects of union leadership on members can be classified into two categories. Under the first, studies argue that instead of a positive impact, union leaders negatively influence members’ job attitudes [1,2], intentionally organizing workers’ dissatisfaction, aimed at encouraging collective action to improve working conditions and social status, resulting in union leaders bringing so-called “merchants of discontent” [3]. Under the second, studies have examined the importance of union leadership on member attitudes, narrowly focusing on union leadership’s effects on member attitudes towards the union, in terms of union commitment, loyalty, and participation [4,5,6,7]. Although there have been significant studies on union leaders, their impact and influence on work attitudes, such as job satisfaction, have been relatively overlooked compared to their importance [8].

To bridge this gap in existing literature, this research is aimed at investigating the effect of union leadership on member job satisfaction, focusing on service-oriented union leadership. Compared to other types of leadership (e.g., transformational union leadership for organizing), service-oriented union leadership is particularly useful in elucidating the satisfaction that union members experience in their workplaces because unions play a fundamental role in serving their interests [9]. Shop stewards’ primary role is to serve members daily by meeting them to handle grievances and inform them about the union’s activities and benefits [10]. Therefore, this study focuses on shop stewards’ role as union leaders and their service-oriented leadership in member job satisfaction.

Besides examining the relationship between service-oriented union leadership and employee job satisfaction, this study also pays attention to union instrumentality as a variable explaining this relationship. Union instrumentality refers to union members’ expectations of the union’s potential role in representing their interests [11]; specifically, it involves union members perceptually assessing whether the union can achieve wage increases, job security, grievance resolution, and other related factors. It can be affected by union leadership [5] and is closely related to job attitudes and performance [12].

However, as union members perform their work in units within the organization, their job satisfaction may be determined not only by union leaders but also by the unit’s managers. In other words, the relationship between service-oriented union leadership and member job satisfaction via union instrumentality may also depend on managers’ leadership styles. In an organized workplace, ethical leadership from a stakeholder perspective can help establish a harmonious labor–management relationship by fostering mutual consideration [13]. Conversely, unethical leadership, characterized by lying, manipulation, and abuse of power, can lead to conflicts with unions, negatively affecting job outcomes. Considering these factors, this study also examines managers’ ethical leadership as a moderating variable in the mediating relationship.

In Section 2, the research develops a research model that includes the hypotheses about the different but synergistic effects of union leadership and managerial leadership on member job satisfaction. These hypotheses are tested with a survey conducted on 850 union members of the electronics manufacturing industry in South Korea, as described in Section 3. In Section 4, this study confirms the positive effect of service-oriented union leadership on member job satisfaction, as well as union instrumentality’s mediating effect and ethical managerial leadership’s moderating effect in the mediated relationship. In Section 5, this study discusses theoretical contributions and practical implications, including union and managerial leadership’s integrative role and union change strategies.

## 2. Literature Review and Hypothesis Development

### 2.1. Union Leadership and Member Job Satisfaction

According to the social information processing theory [14], attitudes are the products of individuals processing information from their surrounding environments. Specifically, this theory suggests that individuals, as adaptive organisms, adapt their attitudes according to their understanding of the surrounding environmental information. In unionized contexts, employees are inclined to adjust their attitudes based on their judgments of union activities or information. As such, union leaders’ (e.g., shop stewards) behaviors may provide union members with social cues and information, affecting their work environment perceptions, and consequently, shaping their attitudes in the workplace [14]. Blanchflower, Bryson, and Green [15] found a significant positive relationship between union membership and job satisfaction among union members in the United States and Europe in a study conducted after the 2000s. Based on the social information processing theory, scholars have recently suggested that union leadership plays a significant role in the formation of union members’ attitudes [16].

The objectives and policies of a union stem from union leaders, influencing member perceptions and attitudes [2,4,5,7,17,18]. Trade unions’ actions affect not only economic and job regulations but also members’ self-fulfillment through their power and authority [19]. Union leaders are the primary providers and key sources of cues and information for these judgments and perceptions [20,21]. General union leaders encourage members to identify strongly with the unions’ collective values and missions [22]. They give meaning to members’ efforts in the collective struggle of voicing their demands by clarifying their efficacy, providing a sense of pride and loyalty, and reminding members of the union’s collective mission [23]. Artz, Blanchflower, and Bryson [24] explained that the positive relationship between union members and job satisfaction is due to minimizing the harm to union members from unemployment.

Effective union leadership affects not only attitudes toward unions but also toward organizations. Previous research has highlighted two viewpoints on union leadership. While one stream has emphasized the effects of union leaders’ transformational leadership [4,6,25], according to which, along with recognizing their members’ existing needs, transformational union leaders also seek to satisfy higher needs, such as collective mobilization and the union’s mission of organizing [23], the other has examined service-oriented leadership [7,26]. Service-oriented leaders are readily accessible to union members, often listening to their concerns, frequently keeping them informed about union activities, and encouraging them to express their opinions [5,7].

This study expects service-oriented union leaders concerned with safeguarding the interests of union members to play an important role in shaping members’ awareness and perceptions of union performance, consequently contributing to improved workplace satisfaction. Service-oriented union leaders are defined as “accessible and available to the employees, listening to their concerns, keeping them informed about union activity, and encouraging expression of opinions” [5]. Service-oriented leadership entails “the daily interaction of leaders with members and perceptions of interest and concern about members” [7]. Note that unions exist to provide a variety of services to their members, and the leaders tend to make decisions to protest and promote member interests on a day-to-day basis [9,27]. Their basic role involves providing members with information on the union’s current activities and plans [5]. Tinuoye, Adamade, and Ogharanduku [28] emphasized the role of trade union leaders in promoting work–life balance, fostering harmonious workplaces, and upholding the dignity of members. Glick, Mirvis, and Harder [29] found positive relationships between overall union satisfaction, including members’ grievance support and job satisfaction. Therefore, this study posits that service-oriented union leadership is likely to positively relate to members’ job satisfaction.

**Hypothesis** **1.**
*Service-oriented union leadership will be positively associated with member job satisfaction.*


### 2.2. The Mediating Role of Union Instrumentality

Prior studies have proved that workers’ cognitive evaluations of unions’ roles affect their attitudes [30,31,32]. In particular, perceived union support and instrumentality have been considered as antecedent factors for not only members’ attitudes, in terms of union loyalty [33] and participation [34], but also organizational attitudes, including turnover intention [35] and job satisfaction [2,36]. This study focuses on union instrumentality rather than union support because it examines employees’ assessments of the economic exchange and tangible benefits associated with their union’s service leadership. Union instrumentality encompasses members’ assessments of the costs and benefits associated with union representation. Hennebert, Fortin-Bergeron, and Doucet [37] demonstrated in a study targeting young workers in Canada that union instrumentality could be strengthened for economic reasons, such as labor conditions, career development, and financial security.

This study expects union leaders’ service-oriented leadership to affect members’ perceptions of union instrumentality. When union members sense that their leader is service-oriented, they assume that he or she is working actively for the good of the union and its members [5]. When union leaders are highly oriented towards promoting union services, and provide sufficient information about union activities through frequent interactions, union members are likely to view the union’s performance favorably. Specifically, if union members are frequently able to meet and talk to union leaders, they may be able to better access information about union activity- and performance-related information. For example, members joining a union for instrumental purposes, such as wage increases, may misunderstand that a wage increase is a favor from the employer, although it may be the result of collective bargaining by the union. Misunderstandings can be corrected through interactions with union leaders, who provide accurate information. The more information a service-oriented union leader gathers about members’ grievances by communicating with them, the better the leader can formulate union policies and plans for bargaining. This study thus postulates a positive relationship between service-oriented union leadership and union members’ perceived union instrumentality.

The research further proposes a positive relationship between union instrumentality, as perceived by union members, and job satisfaction. High-performing unions embrace and uphold individual values and employee rights [38]. In addition, workers’ evaluations of union efforts to obtain desired outcomes, including job security, better working conditions, fair treatment, and a safe workplace through collective bargaining and administrative activities can affect union instrumentality. According to previous research, employees’ expectations of union effectiveness are driven not only by job dissatisfaction [39] but also by their belief that unions can alleviate it and improve their working conditions [11,40,41,42]. This study thus posits that employees with high perceptions of union instrumentality are more likely to have higher job satisfaction than those with low perceptions because they perceive that better working conditions and higher compensations have been made available by their union.

Combining the two, this study anticipates that union instrumentality mediates the relationship between union leaders’ service-oriented leadership and members’ job satisfaction. Thus, we propose the following hypothesis:

**Hypothesis** **2.**
*Union instrumentality will mediate the relationship between service-oriented union leadership and member job satisfaction.*


### 2.3. The Moderating Role of Managerial Ethical Leadership

The social information processing theory regards attitudes as socially constructed realities shaped by social cues. For example, Jabeen, Nadeem, Raziq, and Sajjad [43] argued that support as a social cue from supervisors can create positive attitudes and increase employability. Therefore, if individuals are exposed to social cues conducive to personal satisfaction, they are more likely to experience positive feelings of satisfaction. However, there may be disagreements on whether this situation applies equally to every individual. One significant individual satisfaction-related social cue factor is the relationship with managers. Managers lead with resources, and subordinates tend to follow their leaders as role models through social learning [44,45]. Thus, the relationship between union instrumentality and job satisfaction may be strengthened or mitigated by environmental contexts, such as interrelations with managers or their characteristics.

This study focuses on managers’ ethical leadership in various organizational contexts. Ethical leadership includes open communication, stakeholder perspective, and participation in decision-making because these concepts are important for maximizing cooperative industrial relations’ effectiveness. Ethical leadership is defined as “the demonstration of normatively appropriate conduct through personal actions and interpersonal relationships, and the promotion of such conduct to followers through two-way communication, reinforcement, and decision-making” [46]. Ethical leaders influence their followers’ behavior through appropriate actions that reflect their moral perspective [47,48].

We argue that ethical leadership emphasizes consistent standards of fairness and equity, thereby, influencing work attitudes through interaction with union instrumentality, as perceived by employees. The social learning theory is well-suited for identifying the moderating role of ethical leadership in management and emphasizes that situational factors, such as leaders’ ethical behavior, influence employee motivation, leading to desirable behaviors [44]. For instance, ethical managers encourage responsible business practices through appropriate rewards and punishment [49]. In addition, they are perceived to be people-centered and to care about, develop, and treat employee rights [50]. Most importantly, subordinates’ perceptions of their managers’ ethical leadership increase their willingness to put in extra effort into the job, as well as their satisfaction with the manager [47].

In this study, we propose that ethical leadership espousing workplace justice and fair treatment is synergistic with union instrumentality, which in turn is strongly related to employee satisfaction in the workplace. According to extant literature, ethical managers encourage subordinates to display ethical behaviors in day-to-day relationships. Thus, union members who value union instrumentality and have ethical managers are more likely to have higher job satisfaction because they are motivated to behave positively from multiple sources. By contrast, the positive relationship between union instrumentality and job satisfaction may be undermined by working for unethical managers, as it is more likely that labor–management agreements may not be implemented or that workers’ collective demands may be ignored.

Given that managerial ethical leadership moderates the association between union instrumentality perceived by union members and their job satisfaction, this study posits that it is likely to have a conditional effect on the indirect relationship between service-oriented union leadership and member job satisfaction through union instrumentality, which means that there is an indirect conditional effect pattern among the variables presented in this study. If social information processing and social learning theories are considered integrated, union members’ job satisfaction may be further improved by combining managerial ethical leadership and union instrumentality, enhanced by service-oriented union leadership. As this study predicts a strong (weak) relationship between union instrumentality and job satisfaction when ethical leadership is high (low), we propose the following hypothesis:

**Hypothesis** **3.**
*The strength of the mediated relationship between service-oriented union leadership and member job satisfaction through union instrumentality will depend on managerial ethical leadership. The indirect effect of service-oriented union leadership on member job satisfaction will be stronger when managerial ethical leadership is higher.*


Figure 1 illustrates the research model used in this study.

## 3. Methods

### 3.1. Participants and Procedures

To test the research hypotheses, this study strategically collaborated with two labor union branches to draw samples for data collection. The surveyed union branches are characterized as follows: First, both branches belong to the Metal Workers’ Union and are registered with the relatively militant Korea Confederation of Trade Unions among the two nationwide federations in South Korea. Second, the companies of the two branches belong to the same conglomerate, which had long maintained a non-union policy under the banner of the conglomerate’s founder; however, it has recently recognized unions. Third, both the aforementioned companies are wary of the growth of the unions and have implemented proactive human resource management policies to ensure employee non-grievance.

The survey was conducted online among union members for three months from August to October 2022. In addition to demographic variables, such as gender, age, education level, and tenure, the questions included average wages, service-oriented union leadership, managerial ethical leadership, union instrumentality, and job satisfaction. The data were collected through closed-ended questionnaires (the exact wording and examples of the questions are included in the Measures section). The online survey approach was chosen to facilitate accessibility and convenience for participants. Participants received detailed information outlining this study’s purpose and procedures. Informed consent was obtained through a written consent form before the commencement of the survey. To increase the survey response rate, union officials encouraged participation by emphasizing the purpose of the survey to union members. Also, to ensure confidentiality, all data were anonymized and stored securely. The total number of members in the two branches is approximately 1800, of which 617 participated in the survey, with a response rate of 34.3%. Of these, 603 were finalized after excluding missing data. Of those respondents, 85.3% were affiliated with Branch A, while 14.7% were affiliated with Branch B. The relatively low percentage of respondents from Branch B was due to the smaller number of about 200 union members in Branch B.

Demographically, 87.4% of respondents were male and 12.6% were female. The overall proportion of female members in Branch A was 13.6% (female respondents were 13.8%), while the proportion of female members in Branch B was 5% (female respondents were 4.4%). Therefore, there was not a significant difference in the gender ratio of respondents compared to the overall composition of members. The reason for having more male members seemed to be due to the physical demands of tasks, such as product installation, repair, and technical work. Education-wise, 41.8% and 58.2% had high school diplomas and college degrees or higher, respectively. The average age of the respondents was 40.4 years with 13.2 years of work experience.

### 3.2. Measures

#### 3.2.1. Service-Oriented Union Leadership (α = 0.95)

This study assessed service-oriented union leadership using six items (e.g., “I can contact the union leader easily if I want to”, and “The union leader keeps me well informed about what is going on in the union”) developed by Metochi [7]. The respondents indicated their degree of agreement using a 5-point Likert scale (1 = strongly disagree, 5 = strongly agree).

#### 3.2.2. Union Instrumentality (α = 0.93)

This study assessed union instrumentality using four items from Sinclair and Tetrick [51], who measured traditional union instrumentality in terms of wages, fringe benefits, and job security. Two sample items are as follows: “The union guarantees that employees are fairly rewarded regarding their responsibilities”, and “The union guarantees that employees are protected from unfair treatment”. The survey respondents indicated their degree of agreement using a 5-point Likert scale (1 = strongly disagree, 5 = strongly agree).

#### 3.2.3. Managerial Ethical Leadership (α = 0.95)

Managerial ethical leadership was measured using eight items from the Ethical Leadership Scale developed by Brown, Treviño, and Harrison [46]. Two sample items are as follows: “Manager listens to what employees have to say”, and “Manager makes fair and balanced decisions”. The survey respondents indicated their degree of agreement using a 5-point Likert scale (1 = strongly disagree, 5 = strongly agree).

#### 3.2.4. Member Job Satisfaction (α = 0.73)

Job satisfaction was measured using the five items developed by Spector [52]. The sample items include, “How satisfied are you with your job itself, workload, reward, coworkers, and supervisor?” The survey respondents indicated their degree of agreement using a 5-point Likert scale (1 = not at all satisfied, 5 = highly satisfied).

#### 3.2.5. Control Variables

This study controlled for demographic characteristics known to influence job satisfaction, such as gender (male = 1, female = 2), age (continuous variable ranging from 21 to 60 years), education (dummy coded, high school = 1, college = 2), and tenure (continuous variable ranging from 2 to 38 years). This study also controlled for firm dummy variables and average wages, which may affect job satisfaction.

### 3.3. Analytic Strategy

Structural equation modeling was used to test the mediating effects of union instrumentality and to verify the indirect effect of service-oriented union leadership on member job satisfaction. This study also used PROCESS to specify the conditional effect of union instrumentality on job satisfaction at different levels of managerial ethical leadership using bootstrapping. Bootstrapping was also used to test the conditional indirect effect of union instrumentality and managerial ethical leadership, as recommended by Hayes [53].

## 4. Results

### 4.1. Confirmatory Factor Analysis of Measurement Items

This study performed a series of confirmatory factor analyses to assess the constructive and discriminant validity of our model. First, we ran a four-factor model (service-oriented union leadership, union instrumentality, managerial ethical leadership, and member job satisfaction) to determine how well the model fit the data [54]. For the comparative fit index (CFI) and normed fit index (NFI), values close to 0.95 indicate good model fit [54,55,56]. For the root mean square error of approximation (RMSEA) and root mean square residual (RMR), values less than 0.06 indicate a good model fit and values less than 0.10, an acceptable fit [57]. As seen in Table 1, the four-factor model shows a good fit (χ^2^ = 770.72, *df* = 203, CFI = 0.95, NFI = 0.93, RMSEA = 0.07, RMR = 0.04). Furthermore, we generated alternative three, two, and single-factor models for comparison with the four-factor model. The alternative three-factor model, Model A, subsumed service-oriented union leadership and union instrumentality under one factor; in Model B (two-factor model), service-oriented union leadership, union instrumentality, and managers’ ethical leadership were combined into one factor; and in Model C (single-factor model), service-oriented union leadership, union instrumentality, managers’ ethical leadership, and member job satisfaction were combined into one common factor. However, as seen in Table 1, none of these alternative models yield an acceptable fit. Hence, the research concludes that all four constructs in this study are distinct from one another.

### 4.2. Descriptive Statistical Analysis

Table 2 presents the means, standard deviations, and correlation coefficients of the study variables. Generally, coefficients above 0.70 may increase the likelihood of multicollinearity in a regression [58]. However, all correlations in our study were below this threshold, indicating that all measures are appropriate for inclusion in the analysis, and suggesting that multicollinearity is not a serious problem.

### 4.3. Common Method Bias Test

When data are self-reported, common method bias (CMB) may occur. An important concern in such cases is the artificial inflation of observed relationships among variables. Thus, before testing our hypotheses, this study performed preliminary analyses to assess the potential for CMB. To address this issue, we conducted Harman’s single-factor test [59] by performing an exploratory factor analysis with four substantive variables. CMB is thought to be present when the resulting factor explains more than 50% of the variance [60]. When examining the variables’ unrotated factorial structure, the first factor accounted for 33.15% of the variance, compared to 72.83%, explained by all four factors. These results indicate that a single factor does not account for most of the variance in our data.

### 4.4. Hypothesis Testing

Table 3 presents the results for Hypotheses 1 and 2. Service-oriented union leadership was positively associated with member job satisfaction, as denoted by a significant unstandardized regression coefficient (B = 0.10, *p* < 0.01), indicating that Hypothesis 1 was supported. Following Baron and Kenny’s [61] three-step mediation process, this research analyzed the mediation effect of union instrumentality. The results indicated that service-oriented union leadership was positively associated with member job satisfaction, as predicted in Hypothesis 1, and that it was also significantly associated with union instrumentality (B = 0.69, *p* < 0.01). Additionally, they revealed that union instrumentality was significantly associated with member job satisfaction (B = 0.09, *p* < 0.05), whereas service-oriented union leadership was not significant for member job satisfaction. These results indicated full mediation. Furthermore, this study analyzed 5000 bootstrap samples, constructing 95% bias-corrected confidence intervals for the indirect effect. Table 3 shows that service-oriented union leadership had a significant indirect effect on member job satisfaction through union instrumentality (B = 0.06, CI 95% = 0.01 to 0.11), confirming the indirect effect because the bootstrapped 95% confidence intervals did not include zero, thereby, supporting Hypothesis 2.

Our moderated mediation analysis examined the process by which the dependent variable (member job satisfaction) depends on the value of the moderating variable (managerial ethical leadership). This study examined the moderated mediation by assessing the conditional indirect effect. For the moderated mediation analysis, conditional process modeling was used (Model 14) in the PROCESS macro, as recommended by Hayes [62], which helps to check the robustness of the reported findings. The number of bootstrap samples for the bias-corrected bootstrap confidence intervals was 5000. The strength of the positive relationship between union instrumentality and member job satisfaction differed across managers with low and high levels of ethical leadership. The interaction between union instrumentality and managerial ethical leadership on member job satisfaction was statistically significant (B = 0.09, *p* < 0.01). Conditional process modeling splits the data into low, medium, and high levels of union instrumentality. Table 4 shows that the 95% confidence intervals for the indirect effects at one standard deviation below the mean of managerial ethical leadership included zero (−0.05, 0.07), whereas the mean (0.01, 0.12) and one standard deviation above the mean of managers’ ethical leadership did not (0.04, 0.18), indicating that the conditional indirect effect was strong and significant at high levels of manager ethical leadership but not at low levels, thereby, supporting Hypothesis 3.

## 5. Discussion

Drawing from social information processing and social learning theories, this study has focused on union instrumentality as a cognitive assessment and the mechanism of how service-oriented union leadership impacts member job satisfaction. Furthermore, it has examined whether the strength of the proposed mediation relationship varies according to the extent of managers’ ethical leadership. The analytical results confirm the proposed hypotheses. The theoretical contributions of the findings are as follows: First, this study demonstrates that the social information processing theory, which has been utilized in the study of managerial leadership, can be applied in the context of union shop stewards. The theory suggests that followers interpret the environment and apply it to their behavior through the leader’s actions and information presented through day-to-day interactions [14]. Our findings suggest that in unionized workplaces, union members develop positive expectations about the role of the union through their interactions with union officials, which leads to their job satisfaction. This study has theoretical implications in terms of identifying mechanisms using the social information processing theory, which extends the findings of previous studies showing the direct effect of transformational union leadership on organizational commitment [16].

Second, our research extends the current literature on union leadership, which has primarily focused on union leadership’s impact on union members’ attitudes and behaviors, such as union participation and loyalty [4,5,6,7], by examining union leadership’s influence on job satisfaction. The existing literature on the relationship between union leaders and job satisfaction tends to focus on member dissatisfaction brought about by union leaders to organize collective mobilization. Our study explores the potential of service-oriented union leadership in contributing to job satisfaction by leveraging the representational role of unions in addressing workplace issues. In particular, this study provides clues to interpret these inconsistent and contradictory results, arguing that union members’ job attitudes can be better explained by focusing on union leadership rather than on the membership itself, which means that union members’ job satisfaction is not simply explained by the fact that they are union members, but rather by the way union leaders lead them. The results of this study suggest that the positive relationship between leadership and job satisfaction, as reported in many previous studies [63,64,65], may be replicated in the context of trade unions.

Third, by emphasizing that union stewardship and managerial leadership increase job satisfaction, this study suggests the possibility of an integrated role for industrial relations (IR) and human resource management (HRM) in increasing job satisfaction. The traditional perspective assumes a competitive relationship between the two, suggesting that HRM can replace IR. However, this study demonstrates that union stewards’ service-oriented leadership and managers’ ethical leadership, although distinct, contribute synergistically, increasing job satisfaction among members. Previous studies have yielded conflicting results regarding the relationship between union membership and job satisfaction, leading to lively academic debates [66]. This study contributes to a comprehensive understanding of the relationship between service-oriented union leadership and member job satisfaction by developing and validating a conditional indirect effects model, integrating the instrumental role of unions and the ethical leadership of managers. The focus of this study, labor–management leadership, is connected to research on union-management partnerships that have been traditionally addressed within labor relations and ultimately shares similarities in enhancing organizational effectiveness [67,68].

Fourth, besides influencing job attitudes, service-oriented union leadership can increase the attractiveness of unions to younger generations [69]. This study suggests that service-oriented union leadership may play a more important role than before in the changing employment relationships. As unionization rates are declining, particularly in advanced countries, service-oriented union leadership can enhance the appeal of unions to the younger generation, who prioritize information sharing and transparent union operations. The increasing presence of the Millennial and Generation Z populations in the labor market makes it important to consider their perceptions of the role of labor unions. Previous studies have argued that younger generations consider fair distribution and transparency in unions to be more important compared to previous generations [70]. In this context, utilizing service-oriented union leadership could be a strategic approach to revitalizing trade unions.

This study has practical implications for unions and managements of unionized workplaces. The results highlight the potential for union leaders to contribute to both the union and management. The findings suggest that by developing service-oriented leadership, union leaders can provide maximum support to members, increasing both union instrumentality and job satisfaction. Furthermore, the findings suggest that employers should form and maintain cooperative rather than hostile relationships with unions (including union leaders), considering that service-oriented union leaders contribute to improved job satisfaction among union members. Such cooperation can foster a favorable climate for positive relationships between management and labor and help achieve desirable organizational outcomes for both management and unions. Considering that satisfied employees tend to generate higher value for the organization [71], the findings recommend investing in and fostering service-oriented union leadership to improve job satisfaction, which is bound to yield higher returns.

## 6. Conclusions

This study analyzed the effects of service-oriented union leadership on 603 union members from two branches affiliated with the Korean Metal Workers’ Union. The results of the analysis showed that service-oriented union leadership had a positive impact on job satisfaction, and the union instrumentality was found to mediate the relationship between service-oriented union leadership and job satisfaction. Furthermore, the ethical leadership of managers was confirmed to moderate the mediating relationship, verifying a moderated mediation effect. Although this study was conducted in the Korean context and has limited generalizability, it is impressive that it shows that union and management leadership can enhance organizational effectiveness in Korean metal unions, which are characterized by a rather militant labor movement.

The results of this study should be interpreted in light of the following limitations. First, common concerns about single-source variance apply to the data acquired from union members. Although the research addressed this possibility using Harman’s single-factor test, future research should employ multiple data sources to effectively reduce the possibility of common method variance. Second, as this study used cross-sectional data, we could not guarantee solid causal relationships. Additionally, this study could not control for potential omitted variables, such as union leaders’ tenure and their educational background and experiences, which could have influenced their service-oriented leadership. Future research on union leadership should overcome the present study’s methodological limitations.

For future research, it is necessary to not only overcome the methodological limitations of this study but also examine whether positive attitudes such as organizational citizenship behavior and job performance can be expected, in addition to job satisfaction. Furthermore, considering that union members are not homogeneous, exploring whether the effects of labor and management leadership differ based on factors like generational differences or socio-economic status could help understand the integrated effectiveness of labor and management leadership.

## Figures and Tables

**Figure 1 behavsci-14-00287-f001:**
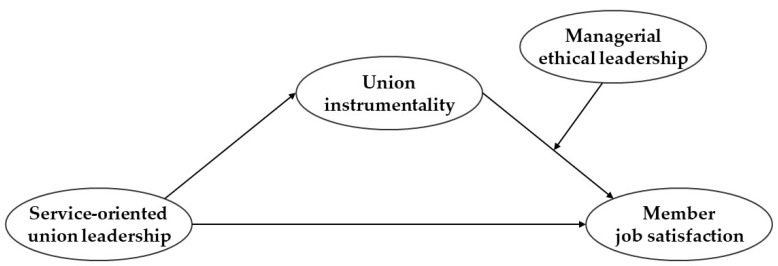
Research model.

**Table 1 behavsci-14-00287-t001:** Comparison of measurement models.

Models	χ^2^ (*df)*	CFI	NFI	RMSEA	RMR
Four-factor model	770.72 (203)	0.95	0.93	0.07	0.04
Three-factor Model A ^a^	1909.68 (206)	0.84	0.82	0.12	0.07
Two-factor Model B ^b^	5305.26 (208)	0.51	0.50	0.20	0.26
Single-factor Model C ^c^	5818.02 (209)	0.46	0.45	0.21	0.27

Note. χ^2^ = chi-squared discrepancy, df = degree of freedom, CFI = comparative fit index, NFI = normed fit index, RMSEA = root mean square error of approximation, RMR = root mean square residual. ^a^ Service-oriented union leadership and union instrumentality combined into a single factor compared to the four-factor model. ^b^ Service-oriented union leadership, union instrumentality, and managerial ethical leadership combined into a single factor compared to the four-factor model. ^c^ Service-oriented union leadership, union instrumentality, managerial ethical leadership, and member job satisfaction combined into a single factor compared to the four-factor model.

**Table 2 behavsci-14-00287-t002:** Means, standard deviations, and correlations among study variables.

	Mean	SD	1	2	3	4	5	6	7	8
1. Gender	0.87	0.33								
2. Age	40.33	7.63	0.27 **							
3. Education	1.58	0.49	−0.07	−0.015 **						
4. Tenure	13.19	7.74	0.24 **	0.63 **	−0.19 **					
5. Average wage	372.97	125.60	0.27 **	0.16 **	−0.05	0.28 **				
6. Service-oriented union leadership	3.76	0.92	−0.09 *	−0.03	0.04	−0.09 *	−0.03			
7. Union instrumentality	3.61	0.95	−0.09 *	−0.08	0.00	−0.08	−0.05	0.66 **		
8. Managerial ethical leadership	2.94	0.86	0.10 *	0.16 **	−0.05	0.09 *	0.00	0.09 *	0.10 *	
9. Member job satisfaction	2.94	0.68	0.07	0.10 *	−0.09 *	0.11 *	0.06	0.12 **	0.14 **	0.42 **

Note. *N* = 603, * *p* < 0.05, ** *p* < 0.01 (two-tailed).

**Table 3 behavsci-14-00287-t003:** Results for direct and indirect effects.

	**Direct Effect**			
	**B**	**SE**	**t**	** *p* **
Service-oriented union leadership → Member job satisfaction (with control variables)	0.10	0.03	3.37	0.00
Service-oriented union leadership → Union instrumentality (with control variables)	0.69	0.03	21.73	0.00
Union instrumentality → Member job satisfaction (with control variables and controlled service-oriented union leadership)	0.09	0.04	2.21	0.03
Service-oriented union leadership → Member job satisfaction (with control variables and controlled union instrumentality)	0.04	0.04	1.06	0.29
	**Indirect Effect**			
	**B**	**Boot SE**	**BCLL**	**BCUL**
Service-oriented union leadership → Union instrumentality → Member job satisfaction (with control variables)	0.06	0.03	0.01	0.11

Note. BCLL = lower level of bias-corrected bootstrap confidence interval, BCUL = upper level of bias-corrected bootstrap confidence interval.

**Table 4 behavsci-14-00287-t004:** Results for moderated mediation effects.

Mediator	Moderator				
Union Instrumentality	Managerial Ethical Leadership	Effect	Boot SE	LL 95% CI	UL 95% CI
Low (−1 SD)	2.07	0.01	0.03	−0.05	0.07
Mean	2.94	0.06	0.03	0.01	0.12
High (+1 SD)	3.80	0.11	0.04	0.04	0.18

Note. Values for quantitative moderators are the mean and plus/minus one SD from the mean.

## Data Availability

The data presented in this study are available upon request from the corresponding author.

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
