# Peer review of "Different but Synergistic Effects of Union and Manager Leadership on Member Job Satisfaction"

_behavsci, 2024, doi:10.3390/bs14040287_

Round 1

Reviewer 1 Report

Comments and Suggestions for Authors

The article is very interesting, and it certainly enriches the existing theory with its content and empirical findings. Although it deals with a topic that can be partly controversial (some authors contrast the activities of trade unions and the pure efficiency of business expressed by the achieved profits), the authors present it with sufficient professional insight and in broader perspectives that are not usually combined in this form.

The premises of the article are fully logical and preferably comprehensible. In this respect, a small imperfection is the length of some sentences (some sentences have 5 or more lines), which makes the text difficult to read. Therefore, it would perhaps be more appropriate to divide (at least some) of the long sentences into several sentences.

The statistical analysis is very precise and gives the article the required scientific erudition and relevance.

Based on the overall positive impression of the submitted article, the authors would deserve a positive attitude from the reviewers as well. Specifically, in an effort to help push the current considerable quality of the article to an even higher level, the following inspirations can be defined:

         The abstract should contain information about the number of involved respondents, which could immediately attract readers with its relevance.

         Perhaps a separate section of the Conclusion could be elaborated, in which the basic contributions of the article would be briefly summarized and, possibly, also defined (at least general) recommendations for improving the situation in the investigated issue. Although the research was conducted in South Korea, the results can be inspiring for other scientific teams around the world, as well as for the top management of organizations and establishing more effective relationships with various forms of employee representatives.

         Perhaps it would be appropriate to incorporate more recent studies into the text of the article (in the current version of the article there are only 2 sources from 2023 and no source from 2022 or 2021) so that the article is "on the pulse of the scientific day".

         In order to preserve higher scientific ambition and its appreciation on the part of the reading professional community, authors should give up personal expressions such as "we" (lines 48, 55, 58, 74, etc.) and replace them with a more formal style (e.g., "the research is aimed to…”).

Good luck.

Reviewer 2 Report

Comments and Suggestions for Authors

In general, the paper is interesting and devoted to an important issue of behavioural economics. The research is well-structured. The empirical part is the strong point of the research. The presentation of the results is coherent and aligned with the hypotheses and method. However, the paper has some drawbacks that should be corrected before acceptance of the article:

1) the important methodological details regarding data collecting are missed - see section 3.1. Particularly, the duration of the survey is hidden as well as the description of the data collection method - was it an online survey? how do the authors involve these 617 respondents and achieve such a high level of responses? the type of the questionnaire and other details of the survey should be added;

2) the gender distribution of the respondents needs more precise justification. It seems it cannot be explained only by R&D specialists' share (who are males generally), but, maybe, it is more connected with the general character of employment in the Metal Workers’ Union. Can it be the reason? So, brief information on the gender structure of the employees is needed - just one sentence, but it is important;

3) the unequal distribution of respondents involved from branches A and B should be explained too;

4) analyzing job satisfaction, the authors omit the overall background of this problem in the literature review. Besides, the majority of the literature used for the theoretical framework of the research is outdated. In this regard, I find it useful to widen the reference list and appropriate parts in section 2 (Literature Review and Hypothesis Development) by recent sources analysis. Valuable information can be found in recent works (Draskovic, V., Pupavac, J., Delibasic, M., & Bilan, S. (2022). Trade unions and hotel industry: Current trends. Journal of International Studies, 15(1), 104-116. doi:10.14254/2071-8330.2022/15-1/7;

Stelmokienė, A., Jarašiūnaitė-Fedosejeva, G., & Kravčenko, K. (2023). Relationship between gender equality, employees‘ perception of distributive justice and wellbeing in EU: The effect of gender and management position. Economics and Sociology, 16(1), 123-137. doi:10.14254/2071-789X.2023/16-1/9; 

Samoliuk, N., Bilan, Y., Mishchuk, H., & Mishchuk, V. (2022). Employer brand: key values influencing the intention to join a company. Management & Marketing. Challenges for the Knowledge Society, 17(1), 61-72. https://doi.org/10.2478/mmcks-2022-0004).

There are many other works related to the area of the research and published since 2021. Please, update the reference list.

Reviewer 3 Report

Comments and Suggestions for Authors

Thank you for a highly interesting paper. I found it precise and balanced in all aspects. It is very well written and clear. I have two minor comments that could develop the article furhter: 

1. The literature is relevant, however, many of the references included are rather old. I urge the authors to review the literature once more in order to include more recent literature. 

2. The discussion section is to the point but rather short and should be elaborated. It only to a small degree interacts with the literature. To position this study to other relevant studies in the field, would benefit the article and provide a more convincing argument. Further, the authors could consider including more contextual information in their discussion to provide more depth. 

Round 2

Reviewer 2 Report

Comments and Suggestions for Authors

Not all notes are considered. Please, revise the previous review carefully.

For instance, I still find the methodological explanation for the survey and data collection poor, particularly, regarding the exact wording of the questions, as well as their type (e.g., open, closed, semi-closed, etc.). All these details are missing. Some missed explanations are added formally like "The reason for the majority of male respondents is partly due to the characteristics of the metal industry and the higher presence of males in repair tasks". What does "higher presence" mean? Don't the authors know the definite number of employed and their structure? It is surprising because the author could achieve written consent from the respondents and had the support of unions in conducting the survey. This explanation should be more precise. 
